# Unusual Canine Distemper Virus Infection in Captive Raccoons (*Procyon lotor*)

**DOI:** 10.3390/v15071536

**Published:** 2023-07-12

**Authors:** Adrian Constantin Stancu, Octavian Sorin Voia, Oana Maria Boldura, Sorin Aurelian Pasca, Iasmina Luca, Anca Sofiana Hulea, Oana Roxana Ivan, Alina Andreea Dragoescu, Bianca Cornelia Lungu, Ioan Hutu

**Affiliations:** 1Faculty of Veterinary Medicine, Horia Cernescu Research Unit, University of Life Sciences “King Michael I”, 300645 Timisoara, Romania; adrianstancu@usab-tm.ro (A.C.S.); oanaboldura@usab-tm.ro (O.M.B.); anca.hulea@usab-tm.ro (A.S.H.); ioan.hutu@fmvt.ro (I.H.); 2Faculty of Animal Resources Bioengineering, University of Life Sciences “King Michael I”, 300645 Timisoara, Romania; 3Faculty of Veterinary Medicine, University of Life Sciences, 700506 Iasi, Romania; spasca@uaiasi.ro; 4Faculty of Letters, West University, 300223 Timisoara, Romania; oana.ivan@e-uvt.ro; 5Faculty of Agriculture, University of Life Sciences “King Michael I”, 300645 Timisoara, Romania; alina.urlica@usvt.ro

**Keywords:** raccoons, canine distemper, *Canine morbillivirus*, RT-qPCR detection, immunohistochemistry, histopathology

## Abstract

Canine morbillivirus, also known as canine distemper virus (CDV), is the causative agent of canine distemper (CD), which is a serious contagious disease of canines, large felids, and, occasionally, raccoons. This study included seven raccoons from the Timisoara Zoological Garden, Romania. CDV was detected using RT-qPCR on blood samples, but several other exams were also performed—clinical, bacteriological, immunohistochemistry (IHC) and histopathology, toxicological screening, and necropsy—which confirmed CDV infection. Severe digestive disorders (diarrhea and frequent hematemesis) were observed. The necropsy findings included pseudo membranous gastroenteritis, congestion, and pulmonary edema in two raccoons. Immunohistochemistry showed immunolabeled CDV antigenantibodies on the viral nucleocapsid. Histopathology revealed lymphocyte depletion in mesenteric lymphnodes and intranuclear and intracytoplasmic inclusions in the enterocytes of the small intestine. Based on the RT-qPCR assay, laboratory tests, and the lesions observed, it was established that the raccoons were infected with CDV, which was the cause of death in two cases. The results from the necropsy, histology, and immunohistochemistry in the raccoons are comparable with reported CDV lesions in dogs. In conclusion, several exams may be performed to establish the etiology of possible interspecific viral infection, but only very specific exams can identify aCDV infection. Laboratory analyses must be completed by RT-qPCR assay or IHC to establish infection with uncommon viruses in raccoons with high accuracy.

## 1. Introduction

Raccoons belong to the family *Procyonidae*, which includes 18 species. *Procyon lotor* lives in a large part of North America, from southern Canada to most of continental North and Central America [1]. In the United States, regular outbreaks occur in free-ranging raccoons (*Procyon lotor*), a species that might play a role in the epidemiology of *Morbillivirus canis*, known as canine distemper virus (CDV) in domestic dogs [2]. In Europe, raccoons are considered neozoa (invasive species), while being used in fur coat production (in German farms). In Romania, raccoons are wild animals sometimes found in zoological parks, but keeping them as pets, albeit illegal, is becoming a new trend.

The infection commonly known as CD is caused by an enveloped single-stranded RNA virus of the genus *Morbillivirus*. It was first isolated by Carré in 1905 [3] and remains a significant concern among veterinarians due to the high morbidity and mortality rates in the animal population [4]. It affects both wild and domestic animals, with dogs being the most common victims [5,6,7,8,9]. CDV has a broad host range [10] as evidenced in several mammalian species: *Canidae* [2,6,11], *Felidae*, *Mustelidae* [12], *Procyonidae*, *Ursidae*, *Viverridae*, and *Hyaenidae*. The infection has also been described in captive and free-ranging large felids [13], *Primates* [10], *Phocidae* [14,15], *Artiodactyla*, and *Proboscidea*. The epidemiology of CDV in raccoons has been described by many authors in recent decades [16,17,18] and in recent studies [1,19].

The primary sources of CDV are infected and carrier animals, which shed large amounts of the virus in all secretions and excretions starting as early as day 5 after infection, before the onset of clinical signs [20,21,22]. Viral shedding may continue for as long as 3 to 4 months but usually resolves after 1 to 2 weeks [22]. Secondary sources include environmental elements that come into contact with pathological products, such as active animated vectors (humans during viremia) or fomites such as water, food, improper transportation and care, contact with bedding or other items infected animals have come into contact with, etc. [5,6,7,8,20,22].

CD is highly contagious and probably transmitted mainly through aerosols [13]. Direct transmission—contact with dead animals or sniffing the soil or vegetation contaminated with viruses from urine or feces—appears to be more probable than indirect transmission. The virus is easily transmitted from domestic to wild animals and vice versa. Any unvaccinated animal is prone to CDV infection, but the most susceptible ones are newborns and animals that live in poor conditions or are fed improperly [15,21].

The incubation period ranges from 3 to 6 days. The virus initially infects lymphoid tissue (monocytes) in the upper respiratory tract and tonsils, subsequently being disseminated via the lymphatics and blood to the entire reticuloendothelial system. In the second stage of cell-associated viremia and fever (8 to 9 days after infection), when CDV infects the cells of the respiratory tract, gastrointestinal tract, central nervous system, urinary tract, and skin, as well as white blood cells [22].

The risks are lower if the infection is diagnosed early and if the immune status of an animal is in good condition. The only way to prevent infection in dogs and non-domestic captive hosts is to vaccinate them, according to routine practice [23,24,25]. Many authors have suggested the possibility of cross-species infection from wild animals to zoo animals or from domestic to wild or captive animals because of CDV virulence and the limited preventive or therapeutic approaches in such hosts [6,7,8,9,26,27,28].

This study was performed to establish the cause of infection in seven raccoons from Timisoara Zoological Garden, Romania, and the cause of death in two raccoons. Ancillary laboratory tests and necropsies were performed to verify the hypothesis that infection and death were caused by a possible interspecific infection with CDV.

## 2. Material and Methods

The study started upon identifying the first clinical signs and included the following: necropsy, laboratory analyses (RT-qPCR, toxicological and microbiological screenings, histopathology and immunohistochemistry, electronmicroscopy) to identify the etiological agent. Other exams would follow upon approval of Ethical Statement no. 73/2020; however, the animals had meanwhile been displaced from the Zoo.

### 2.1. History, Examination, and Preliminary Treatment

The seven raccoons from the Timisoara Zoological Garden, Romania lived in a separate area, isolated from other domestic or wild animals while being visible to visitors. The Zoo veterinarian reported that two out of seven raccoons, aged 7 months, showed clinical signs of digestive system disorders, characterized by severe diarrhea and hematemesis, which are common in CDV infection, but also in other microbiological infections or other causes. No clinical signs were detected in five of the animals. None of the raccoons had been vaccinated against CDV, as the infection had not been reported in zoological parks either in raccoons or in other susceptible species in this type of habitat. The blood samples were collected in EDTA vacuum blood collection tubes at the beginning of antibiotic treatment and transported to the laboratory for screening in an ice box at approx. 4 °C. Although the veterinarian isolated the animals in individual cages and attempted treatment with Enrofloxacin (5 mg/kg body weight) and lactated Ringer solution for rehydration (3 mL/kg and hour) twice a day, two raccoons died 2 and 3 days after the onset of clinical signs.

### 2.2. Detection of CDV by RT-qPCR

CDV was detected in the raccoons by RT-qPCR assay. Total genome extraction was performed from blood samples using a QIAamp Viral RNA Kit (Qiagen, Hilden, Germany) following the manufacturer’s data sheet protocol. Viral antigen was detected using the IVD virus detection kit for veterinary practice One-Step Distemper (Bioingentech Biotechnologies, Concepción, Chile). The qualitative real-time PCR assay was performed using an Mx3005P real-time PCR probe-based technology (Agilent Technologies, Santa Clara, CA, USA), according to the data sheet protocol. Each sample was analyzed in triplicate: a negative control, a positive control, and an internal reaction control were used for each run. The results were interpreted following the kit producer’s data sheet protocol according to Ct values, as follows: a value lower than 11 and higher than 40 was considered negative, Ct values between 12 and 35 were considered positive, and values higher than 35 and lower than 40 were considered inconclusive.

### 2.3. Necropsy

The raccoon cadavers were transported under biosecure conditions in sealed plastic bags from the Timisoara Zoo to the Laboratory of Pathological Anatomy within the Faculty of Veterinary Medicine in Timisoara to establish the cause of death. The necropsy was performed according to mammalian autopsy protocol after skinning the carcasses, opening the thoracic and abdominal cavities, and examining the tissues and organs, looking for macroscopically visible lesions.

### 2.4. Toxicology

Since the toxicological examination required a short turnaround time, it preceded the PCR test. The purpose of this rapid intervention was to ensure the survival of the remaining raccoons, in case intoxication/poisoning was the cause of the clinical signs and deaths. To exclude the possibility of intoxication/poisoning, prior to obtaining the results from RT-qPCR, and after the necropsy, the liver samples were analyzed to identify the main anticoagulant raticides on the Romanian market (difenacoum, brodifacoum, bromadiolone, and warfarin). The technique used for this purpose involved high-performance liquid chromatography (HPLC). An Agilent HPLC 1100 was used to analyze the anticoagulant rodenticides with a C18 analytical column, 300 mm × 4.5 mm, 5 µm particle size, and a 5 µm guard column. The analysis was performed using 50 mM of sodium acetate (pH 6.5) as solvent A and methanol as solvent B at a constant flow rate of 1 mL/min in a gradient run. Linear gradient elution was used by means of solvent B from 30% to 65% over 0–17 min, followed by linear gradient elution of solvent B from 65% to 90% over 17–24 min. The column was washed with 90% solvent B over 5 min, followed by 4 min equilibration to the initial conditions. The fluorescence detector was set to 318 nm excitation and 400 nm emission, and the excitation spectrum was recorded at 240–360 nm. The injection volume was 30 µL and the column temperature was adjusted to 25 °C.

### 2.5. Microbiology

To exclude *E. coli* bacterial infections, samples from the small intestine (duodenal and jejunal segments), liver, and lymph nodes were analyzed to isolate the bacterial strains. The samples were incubated in nutrient broth for 12 h at 37 °C. Subsequently, the primary cultures were cultured on TBX medium (tryptone bile agar, for 24 h at 44.5 °C) and EMB agar (eosin methylene blue agar, for 24 h at 37 °C). The colonies showing characteristic figures on TBX and EMB agar were subjected to biochemical tests (indole, methyl red, catalase, citrate, Voges–Proskauer) for confirmation. The hemolytic activity was evaluated on Columbia agar. Biofilm formation as a virulence factor was detected by the Congo red agar method.

Sabouraud glucose agar gel (Beckton Dickinson GmbH, Heidelberg, Germany) was used for mycological examination with chloramphenicol and gentamicin (50 mg of each antibiotic/1000 mL). Inoculation was performed in aerobic conditions for 48 h at 37 °C. Any grown fungal colony was subjected to methylene blue staining and lactophenol cotton blue staining. The morphological characteristics of the isolated strains on smears stained with methylene blue revealed the presence of *Candida* spp.

### 2.6. Histopathology and Immunohistochemistry

For the histopathology, samples were collected from the lung and intestine to highlight the inclusions produced by the virus. The techniques applied in staining intestinal and lung samples included fixation (formalin), embedding (dehydration, clarification, fixation, and inclusion in paraffin), sectioning (6 µm with the Slee Mainz microtome), and hematoxylin and eosin staining (H&E stain). Tissue was collected from each intestinal portion for the tissue processing protocol. Each sample was fixed in 4% paraformaldehyde for 24 h, after which the samples were washed in tap water and kept in successively increasing volumes of alcohol solutions. The samples were then embedded in paraffin blocks, which were sectioned using a (4 µm thick) microtome and placed on glass slides. The samples were subjected to paraffin removal, rehydration, washing, neutralizing endogenous peroxidase, protein block incubation, washing with TBS 1, and the addition of a conjugate consisting of the primary antibody coupled with peroxidase in a dilution of 1:100 for each sample. After being kept for 12 h in trays with water in the refrigerator, the slides were subjected to the following: washing with TBS 1, incubation with post-primary antibody, washing with TBS 1 and incubation with Novolink polymer containing the primary antibody for 30 min, rinsing with TBS 1, treatment with 3,3′-Diaminobenzidine for 5 min, and washing with distilled water. Subsequently, hematoxylin staining of the nuclei was performed for 40 s. Finally, the slides were washed with distilled water (two successive water baths for 5 min), followed by afinal washing of the slides with Unyhol, Unyhol Plus, and BioClear.

### 2.7. Immunohistochemistry and Electron Microscopy

Immunohistochemistry enabled the fixation of the primary antibodies coupled with peroxidase by the viral nucleocapsid. For this purpose, portions of intestines with macroscopic lesions were collected from the carcasses. The paraffin-embedded sections were deparaffinized, rehydrated, and incubated with the primary antibody (1:100 dilution, CDV Antibody DV2-12 NB100-64816, Novus Biologicals, USA) overnight at 4 °C. Primary antibody labeling was performed using a polymer detection system (Novolink max Polymer detection system, Novocastra Leica Biosystems) and DAB (3,3′-diaminobenzidine, Novocastra Leica Biosystems) as a chromogenic substrate. Hematoxylin staining was also performed prior to dehydration and mounting. For the negative control, the primary antibody was substituted. Images were acquired with aCX 41 microscope with a 3MP CMOS digital color camera (Olympus).

The transmission electron microscopy (TEM) investigation was performed with a Hitachi-7100 microscope (Hitachi High Technologies, Ibaraki, Japan) at 75 kV. Viral particles were visualized in a suspension (sample) according to their specific adsorption on the surface of a double-membrane electrolytic network (formvar and carbon), followed by fixation, washing, and negative staining (deposition of electron-dense substances around viral particles) by transmission electron microscopy. For the electron microscopy (EM-DNSM) by direct negative staining, lung and intestine segments were placed in a suspension of quartz sand (Merck, KGaA, Darmstadt, Germany) and phosphate-buffered saline (PBS, 5 mL at pH 7.2–7.4) in sterile conditions. The suspension was harvested and subjected to clarification by centrifugation at 400× *g* for 20 min at +40 °C. An amount of 50 µL of suspension was taken from the resulting supernatant, covered with 150-mesh electrolytic grids (copper mesh grids) with a double membrane (formvar and carbon) for 1–2 min. The grids were contrasted with 2% uranyl acetate in distilled water, followed by an examination by electron microscope. The lung and intestine segments collected were fixed for 30 min in cold water in PBS, with 2.5% glutaraldehyde. Subsequently, the parts were post-fixed in osmium tetroxide (OsO_4_) solution, dehydrated by successive passages in ethyl alcohol baths in increasing concentrations, followed by propylene oxide clarification, and included in Epon 812. Ultrafine preparations sectioned by LKB III ultramicrotome were placed on electrolytic grids and double contrasted in Reynolds solution. 

## 3. Results

### 3.1. Clinical Outcome

The five raccoons did not show any clinical signs, suggesting that the infection hadnot affected all the animals. After treatment, both dead raccoons displaying clinical signs were in good body condition [29] before death but displayed lower bodycondition scores and a delay in development as compared to the other animals. 

### 3.2. Detection of CDV by RT-qPCR

Following analysis by RT-qPCR [30,31], the viral genome was detected in all seven blood samples collected. The results, including those of the dead animals, are presented in Table 1.

### 3.3. Necropsy

Gross pathology was characterized by hemorrhagic gastroenteritis (Figure 1a), pulmonary congestion (Figure 1b), pharyngeal ulcers, and foci of necrosis on the liver and pancreas surface. The severity of detected anatomo-pathology and clinical signs in the hosts was associated with the presence of secondary infections with *E. coli*. An examination of the small intestine revealed nematodes of the genus *Ascaris* spp. Most probably, the death was caused by hypovolemic shock.

### 3.4. Toxicology

The HPLC results were negative. For the linearity assessment parameters (with R^2^ = 0.997), the limits of detection (LOD) and limits of quantification (LOQ) obtained from blank samples spiked with anticoagulant rodenticides were LOD = 0.003 mg/kg and LOQ = 0.010 mg/kg; with difenacoum, brodifacoum, and bromadiolone, LOD = 0.010 mg/kg and LOQ = 0.030 mg/kg with warfarin. No detectable anticoagulant rodenticides were found in samples over the LODs, and thus intoxication/poisoning was excluded.

### 3.5. Microbiology

During the microbiology, no potentially pathogenic *E. coli* associated with the lesions were identified. The *Candida* spp. was identified by mycological examination of samples taken from the intestine. 

### 3.6. Histopathology and Immunohistochemistry

The histopathology of lung samples displayed active congestion and characteristic modifications of fibrinous bronchopneumonia. Intestine mucosa showed diffuse necrosis localized in the mucosa (Figure 2a), and both intracytoplasmic and intranuclear inclusions (Figure 2b,c). Additionally, even if no clinical signs were reported, encephalic cell histopathology displayed negative results. Inclusions at the level of encephalic cells and characteristic lesions occur very rarely in adenovirus infections, as in the case under discussion.

### 3.7. Immunohistochemistry and Electron Microscope Examination

The immunohistochemistry of the small intestine showed immunopositivity, as seen in the germinal centers of the medullary area (Figure 2d). This confirms the presence of CDV, as well as viral antigens in the cytoplasm of enterocytes. Inclusions have been reported by some researchers [21,22], while they have not been shown in raccoons previously. Thus, to confirm the etiological diagnosis with certainty, electron microscopy was utilized.

Following this examination, the ultrastructural aspects of the enterocytes were highlighted: electron-dense, enveloped agglomerations having a spherical geometry, a diameter of about 150 nm, characteristic of morbilliviruses that replicate in the cytoplasm by attaching the virus to the host cell (Figure 3).

Electron microscopy revealed viral particles characteristic of morbilliviruses in lung samples.

## 4. Discussion

In our case, the infections may have been caused by secondary sources of infection through active animated vectors (free-ranging small carnivores such as mustelids, or animal caregivers during the period of viremia) or abiotic fomites (water, food, means of transport and care, etc.). This may be the case, given that the zoo is located in a wooded area on the outskirts of the city of Timisoara, and many stray dogs can reach the area; (unfortunately, the problem of stray dogs in Timisoara is not yet fully resolved). Unvaccinated dogs have a high prevalence of CDV infection (according to unpublished data from a retrospective unpublished study) and they may be sources of infection. Moreover, the raccoons in our study could have come into contact with animal lovers (visitors, caretakers, etc.) who later entered the zoo and came into contact with them. 

One limitation of the study is due to several factors pertaining to the specificity of the research conducted. The intention was to later take other samples from the raccoons that survived to detect CDV-specific antibodies after convalescence, but while waiting for the Ethical Statement to be issued, the zoo in Timisoara, Romania was closed down by the local authorities, and the animals were relocated. Therefore, the RT-qPCR data reported in Table 1 remain the sole evidence that the five surviving raccoons were also infected with CDV.

The clinical signs of CD vary dramatically and are highly dependent on the virus strain, age, and immune status of the host. A serological survey has proved that many free-ranging carnivores, puppies, and dogs experience subclinical infection, followed by death [6]. The virus replicates in epithelial cells of the gastrointestinal tract and/or respiratory system. Gastrointestinal tract or respiratory signs may include fever, bilateral serous and nasal ocular discharge, conjunctivitis, and non-productive cough. Secondary bacterial infection with virus-induced immunosuppression may lead to the development of mucopurulent nasal and ocular discharges and bacterial bronchopneumonia, with tachypnea, productive cough, lethargy, and decreased appetite. Epithelial erosion of the gastrointestinal tract caused by the virus may result in inappetence, vomiting, diarrhea, electrolyte imbalance, and dehydration. Dogs that mount an intermediate or delayed immune response may recover from acute illness but fail to eliminate the virus completely, which leads to a spectrum of chronic infection manifestations, which often involve the uvea, lymphoid organs, footpads, tooth enamel, and especially the central nervous system (CNS), with up to 30% of infected dogs developing CNS signs [6,22]. The severity of CD clinical signs depends on strain virulence, environmental conditions, host age [32], and immune status in dogs [33] and raccoons. The clinical signs and lesions in infections with CDV may include respiratory, digestive, cutaneous, and nervous issues. The clinical signs found in raccoons were similar to those found in dogs [8]—the animal species that is most commonly affected by this infection—and they included anorexia, indigestion, vomiting, diarrhea, and hematemesis. It is likely they were deprived of the proteins required for antibody synthesis due to anorexia, which could have affected their immune system. Given that death occurred after 2 and 3 days, respectively, no clinical signs of nervousness were reported, as these usually set in approximately 14 days after the onset of the infection [34]. As others report [35], the infection caused by CDV may range from subclinical to severe, and it is sometimes fatal.

There is no specific treatment to cure this infection, but there are methods of treating clinical signs. Depending on the severity of the condition and how advanced the infection is, IV fluid infusions are recommended if the animal is dehydrated and antimicrobial drugs are recommended for secondary bacterial pneumonia, as well as oxygen supplementation and nebulization [22]. The response to treatment of CDV clinical signs may vary, depending on the size of the animals, their health, age, and the degree of infection. To prevent infection, CDV vaccines must be given at the age of two months with a booster administered after two weeks [32,35,36] in both dogs and raccoons, or every 2–4 weeks until the puppy is at least 16 weeks of age in raccoons [37].

CDV in raccoons may be tested with nasal swabs, as well as PCR on tissue samples, as it is done in dogs. Based on our experience with PCR analysis, an ideal sample type varies based on clinical signs. A PCR exam is recommended after observing the following clinical signs: gastrointestinal signs (whole blood and/or feces), respiratory signs (nasal, pharyngeal, or ocular swabs), or neurologic signs (whole blood, urine, and/or conjunctival swabs). RT-qPCR can be used to detect CD in raccoons, as it is a high-sensitivity (98.9%) and high-specificity (100%) method used in CDV diagnosis in dogs [30,31,38]. Detection in blood samples is common and easy in dogs [30,31], while detection in urine and feces samples from raccoons may be more demanding, despite having easier access in zoological gardens. Other authors include comparisons of serum antibody titers to CDV in the cerebrospinal fluid (CSF) [22]. However, collecting CSF is an even more challenging task. Gross pathology was characterized by severe dehydration due to severe enteric phenomena, pulmonary congestion, and hemorrhagic enteritis, as reported by other researchers [24].

The presence of bacteria and fungi has also been reported as secondary bacterial infections [39,40]. As in dogs, the presence of bacteria and fungi may cause death in raccoons. CDV infection of lymphoid cells leads to immunosuppression, the severity of which may lead to variability in the clinical infection with the potential for viral co-infections with coronavirus (CCoV) [41], adenoviruses (CadVs) [42], herpesvirus (CaHV-1) [43], rotavirus (RVA) [44], parvovirus (CPV2) [45], or secondary nonviral co-infections like bacterial infection (for this reason the antibiotic treatment is used). It may also eventually lead to the development of neurological signs in its later stage [46]. Additionally, coccidian parasites, neospora, tenia, or nematodes have been identified [47] as non-viral co-infestations with CDV. In our case, the examination of the small intestine revealed nematodes of the genus *Ascaris* spp., which may explain the eosinophils detected by the histopathology.

The histopathology characteristic of CDV infection, namely intracytoplasmic and intranuclear inclusions, correspond to those mentioned by other researchers in raccoons [25] as well as in dogs [48]. Since the lesions of the digestive tract resembled those caused by parvovirus, namely the necrosis of the glandular crypts (Figure 2a), immunohistochemistry was performed to exclude it. However, the results were negative, as they lacked the brown coloration characteristic of the antigen–antibody reaction in the case of positive parvovirus results. The results obtained by the IHC technique are similar to the results found in the literature on the use of this method in the diagnosis of CDV [49,50], for instance in dogs [48]. Moreover, the ultrastructural aspects of enterocytes observed by electron microscopic examination were also reported by Habermann et al. [32].

As stated in the literature, during the evolution of CDV, immunity played an important role in triggering the infection and causing its severe evolution. Due to the high morbidity and mortality rates and broad host range (domestic dogs and wildlife, in at least 6 orders and over 20 families of mammals), understanding the epidemiology of CDV is important for its control in both domestic animals and wildlife [51,52,53]. From an epidemiological point of view, the existence of CD in raccoons from the Timisoara Zoo, Romania increases the risk of transmission to pets in the Timisoara area. Uncontrolled imports [54], and unvaccinated or improperly vaccinated dogs between 3 and 6 months of age are at higher risk of infection [38,55]. As reported in the literature [38,51,52,56] and as seen in this outbreak of CDV, infection is clearly possible, and the epidemiological risk of infection transmitted by raccoons to dogs is likely to occur through secondary sources like bedding or other objects touched by infected animals.

Although CDV is not believed to cause disease in humans, two decades ago, it played a controversial role in Paget’s bone disease [57]. CDV RNA has previously been detected in lesions, but other studies have failed to detect CDV nucleic acid in lesions, probably due to contamination. CDV has been shown to replicate in human osteoclast precursors, further raising concerns about the possibility of zoonotic transmission of CDV [58]. Systemic CDV infection resembling CD in domestic dogs may also be found in wild canids, procyonids, ailurids, ursids, mustelids, viverrids, hyaenids, large felids, or seals. The broad and expanding host range of CDV and its perpetuation in wildlife reservoir hosts considerably hampers disease eradication [59]. Thus, considering that CDV may occur in domestic or wild species without any clinical signs (asymptomatic), given the neozoa characteristics of raccoons (as an invasive species in Europe [60]), and in view of the risk of transmission to other raccoons, domestic animals, or humans, it may be concluded that vaccination is crucial for the prevention of this disease [28] and high accuracy diagnostic methods have to be used to identify CDV as soon as possible upon the onset of clinical signs.

## 5. Conclusions

As seen above, the clinical signs (severe diarrhea leading to severe dehydration and eventually to death from hypovolemic shock), as well as the histopathology (including immunohistochemistry) and electron microscopy images obtained in raccoons are comparable with observations reported in dogs [32,48]. Based on the immunohistochemistry of the intestine, the presence of viral antigens in the cytoplasm of enterocytes (inclusions produced by morbilliviruses) may be considered a confirmation of CDV in raccoons but high-accuracy diagnosis requires RT-qPCR examinations of blood or tissue samples, or nasal swabs from suspected infected raccoons.

In conclusion, zoological parks may help reduce CVD outbreaks by minimizing contact with infected animals, prohibitingvisitors’ pets, controlling wild host animals, and vaccinating captive animals [61], which may be carriers of CDV.

## Figures and Tables

**Figure 1 viruses-15-01536-f001:**
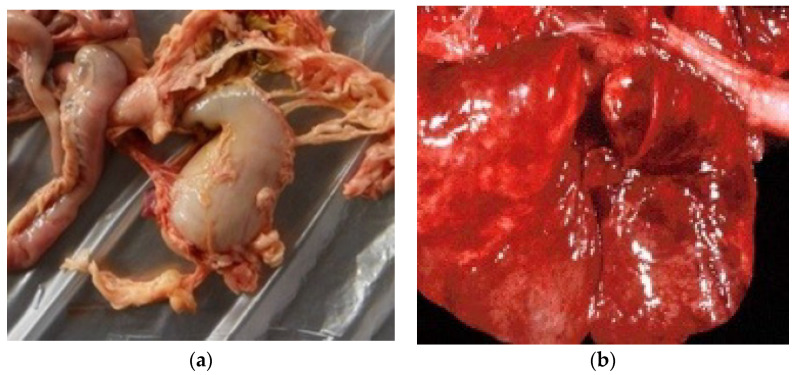
Macroscopic views of the large intestine and lungs. (**a**) Macroscopic large intestine lesions: enteritis, hyperemia, increased volume, and weight; (**b**) Acute lung infection—congestion and active congestion in bright red areas and bronchopneumonia in burgundy red areas.

**Figure 2 viruses-15-01536-f002:**
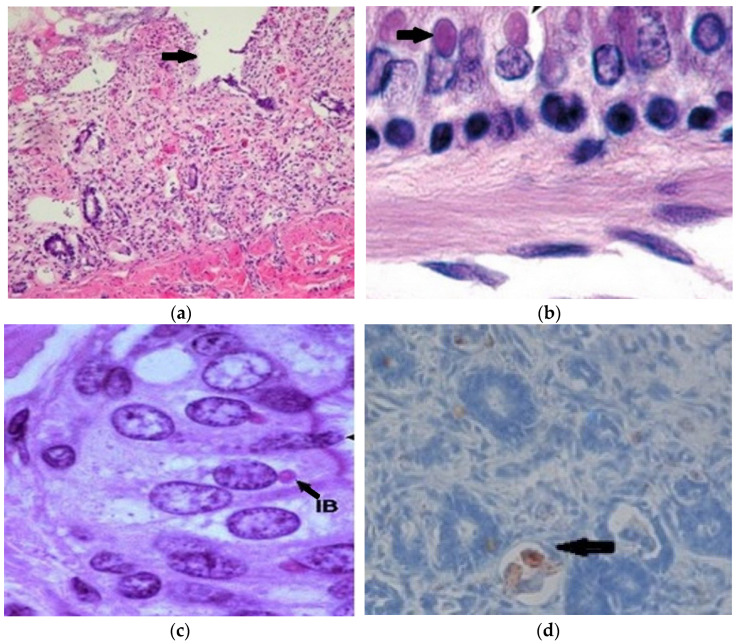
Histopathology and immunohistochemistry of the small intestine. (**a**) Necrosis/loss of intestinal crypts (HEx10); (**b**) cytoplasmic eosinophilic inclusion bodies (HEx10); (**c**) cytoplasmic eosinophilic and intranuclear inclusion bodies in enterocytes (HEx20); (**d**) immune positivity in mononuclear inflammatory cells (IHCx10).

**Figure 3 viruses-15-01536-f003:**
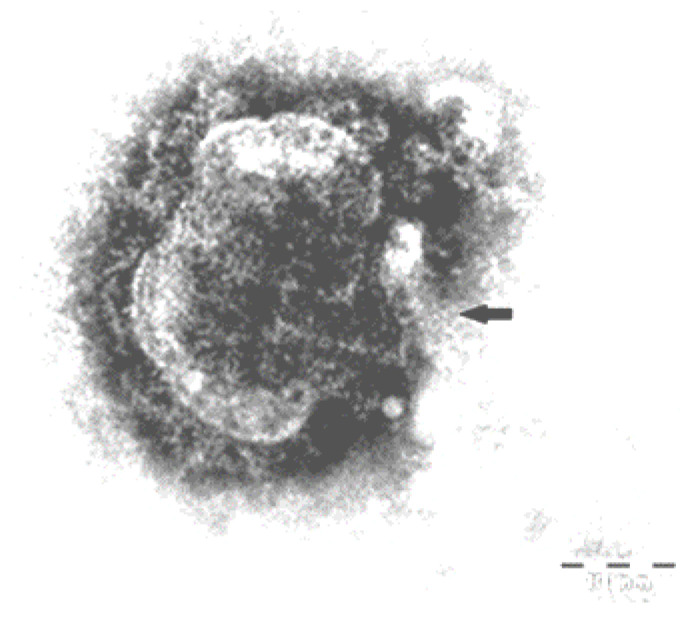
The intestine. Ultrastructural appearance of an enterocyte, showing electron-dense agglomerations of intracytoplasmic and intranuclear morbilliviruses. Negative staining, ×60,000.

**Table 1 viruses-15-01536-t001:** RT-qPCR results from blood samples.

Sample Code	Clinical Signs	Ct Value ^1^	Results
1	Healthy, no clinical signs	33.8	positive
2	Healthy, no clinical signs	31.4	positive
3	Healthy, no clinical signs	32.6	positive
4	Healthy, no clinical signs	34.5	positive
5	Healthy, no clinical signs	32.7	positive
6	Clinical signs: diarrhea, hematemesis, death	19.4	positive
7	Clinical signs: severe diarrhea, hematemesis, dehydration, death	21.2	positive

^1^ Ct values between 12 and 35 were considered positive.

## Data Availability

All of the data presented were obtained from subjects involved in this study. Data availability statements are available.

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
