# Peer review of "Unusual Canine Distemper Virus Infection in Captive Raccoons (Procyon lotor)"

_viruses, 2023, doi:10.3390/v15071536_

Round 1
Reviewer 1 Report
This manuscript by Stancu et al. has one major findings:
1- The raccoons were infected with CDV. These animals are confined in the Zoo and possibly do not have direct contact with other animals.
The authors' idea was interesting, but as highlighted below, there is no in-depth analysis of Morbillivirus canis-positive selected raccoons. The results presented are not expressive for the epidemiology of Morbillivirus canis. Thus, in my opinion the data in the current state unsatisfactory in terms of scientific content and the study does not meet the merits for publication in Viruses.
Major comments
1. According to the ICTV nomenclature, the correct name is Morbillivirus canis. I see it important to keep this taxonomy in the title. In the summary and introduction, if you want to keep Canine distemper virus, I suggest the following: Morbillivirus canis (previously known as canine distemper virus (CDV)).
2. I see no need for “Simple Summary” section.
3. Line 17 and 19: maintain the use of nomenclature (CDV) mentioned above.
4. Line 40: Add more information in the keywords item.
5. Line 54: Please considerer writing It affects instead of Itaffect.
6. Line 63: Consider properly separating the words (i.e., recoveredanimals, whichshed...).
7. Consider grouping the sentence from line 70 with the paragraph from lines 71-75.
8. Lines 105-106: What is the process number approved by the University Ethics Committee?
9. Materials and Methods: The methodology followed by the authors is not clear. For example: in item 2.2 the qRT-PCR reaction is not clear, even without citing primers, probe/SYBR or reference. The reference is only mentioned in another line already in the result section (lines 245 and 246). The methodology must be transparent to be reproducible by other authors. Another example: 2.6 "A kit containing a specific immunoglobulin conjugate..." what is the specification of this kit? Therefore, reformulate the Methodology items.
10. In item 3.4 (lines 255 to 258) I see it important to add more information about this in a supplementary file.
11. Figures 1 and 2: Improve the description of the relevant data of the photos.
12. 3.7 The electron-microscopic examination: I don't see Figures 3 to 6 as electron microscopy results.
13. Discussion: Lack of discussion about the limitations of the study. The discussion also does not address the potential impact of the findings on public health or the implications for the transmission of CDV from raccoons to other species.
Author Response
1- The raccoons were infected with CDV. These animals are confined in the Zoo and possibly do not have direct contact with other animals.
The authors' idea was interesting, but as highlighted below, there is no in-depth analysis of Morbillivirus canis-positive selected raccoons. The results presented are not expressive for the epidemiology of Morbillivirus canis. Thus, in my opinion the data is in the current state unsatisfactory in terms of scientific content and the study does not meet the merits for publication in Viruses.
We hope that after all corrections were introduced, the value of the paper has largely been improved in accordance with all reviewers’ suggestions, thereby being publishable in the journal.We believe that, even if the case in point may not be expressive for the general epidemiology of Morbillivirus canis, it is still worthy of being considered under the guise of a case study on [viral variation patterns within] a limited sample target. To improve the relevance, we have also included more in-depth analysis of Morbillivirus canis-positive selected raccoons.
Major comments
- According to the ICTV nomenclature, the correct name is Morbillivirus canis. I see it important to keep this taxonomy in the title. In the summary and introduction, if you want to keep Canine distemper virus, I suggest the following: Morbillivirus canis (previously known as canine distemper virus (CDV)).
We introduced the suggestion – although it isno longer proper to call the disease CDV,it is more suggestive and familiar forveterinarians/infectious diseasepractitioners.
- I see no need for “Simple Summary” section.
We deletedthe “Simple summary”.
- Line 17 and 19: maintain the use of nomenclature (CDV) mentioned above.
We maintained the CDV acronym for veterinarians/practitioners’ better understanding.
- Line 40: Add more information in the keywords item.
We added more keywords: Raccoons, Morbillivirus canis, Canine distemper virus, qRT-PCR detection, Immunohistochemical exam, Histopathological exam
- Line 54: Please considerer writing It affects instead of Itaffect.
We accepted the correction.
- Line 63: Consider properly separating the words (i.e., recoveredanimals, whichshed...).
We accepted the correction (they actually appear as separate words in our initial document).
- Consider grouping the sentence from line 70 with the paragraph from lines 71-75.
We accepted the correction.
- Lines 105-106: What is the process number approved by the University Ethics Committee?
In the Institutional Review Board Statement paragraph: The animal study protocol was approved through Ethical Statement no 73/2020.
- Materials and Methods: The methodology followed by the authors is not clear. For example: in item 2.2 the qRT-PCR reaction is not clear, even without citing primers, probe/SYBR or reference. The reference is only mentioned in another line already in the result section (lines 245 and 246). The methodology must be transparent to be reproducible by other authors. Another example: 2.6 "A kit containing a specific immunoglobulin conjugate..." what is the specification of this kit? Therefore, reformulate the Methodology items.
Answer from the Genetics lab:
Thank you for your valuable comments. For qRT-PCR analysis, a detection kit marketed by Bioingentech Biotechnologies was used. This IVD kit is used in veterinary practice and is not intended for research. Unfortunately, such a kit does not provide much information about the method or technology used. There is no information about the primer pairs used nor about the technology. According to the filters that were set (VIC, FAM) for the qPCR method, it can be deduced that the kit is based on a probe-based technology.
Answer from the Genetics lab: Also, the way of interpreting the result of the analysis, "a value lower than 11 and higher than 40 was considered negative, Ct values between 12 and 35 were considered positive, and values higher than 35 and lower than 40 were considered inconclusive" was taken with accuracy according to the instructions of the kit.
In the instructions of the kit, it is specified that an RNA sample is needed, which is inserted into the ready-to-use reaction mixture, as well as the program and the PCR method to be used.
According to your requirements, we have reformulated the section to make it more straightforward for the readers.
- In item 3.4 (lines 255 to 258) I see it important to add more information about this in a supplementary file.
We have added more details about the HPLC system and the results even though they were negative - under the limits of detection of the equipment.
- Figures 1 and 2: Improve the description of the relevant data of the photos.
We added more relevant information.
- 3.7 The electron-microscopic examination: I don't see Figures 3 to 6 as electron microscopy results.
Figures 3-6 indicate the results of histopathological examination. Just figure 7 indicateselectron-microscopy results. We moved figures 3-6 up.
- Discussion: Lack of discussion about the limitations of the study. The discussion also does not address the potential impact of the findings on public health or theimplications for the transmission of CDV from raccoons to other species.
We now discussedthe limitation of the study and we also included the public health perspective and implications for the transmission to other species.

Reviewer 2 Report
The authors describe an outbreak of canine distemper virus among raccoons in the Timisoara Zoo, Romania. Although the study contains some interesting observations, the manuscript needs revision before it can be considered for publication.
1. The title should be rephrased: this study does not describe interspecies (or interspecific) transmission. The source of the virus remains unclear. Moreover, the outbreak was in raccoons in a zoo, not in wild raccoons. In addition, the word “similarities” is misplaced, and should not be used in the title.
2. Line 15, 25 and 38: CDV does not cause disease in domestic cats, only in large felids.
3. Line 17 and 27: qRT-PCR using which clinical samples? This should be mentioned in summary and abstract.
4. Line 21/22 and 37: there are NO data on interspecific infection in this study.
5. Line 48-50: there are large numbers of free-ranging raccoons in Europe, see e.g. https://www.mdpi.com/2068982
6. Line 70-72: there is no evidence that CDV is PRIMARILY transmitted by aerosols. Between carnivores and omnivores the direct and indirect contact route (contact with dead animals or sniffing urine or feces) is probably also important.
7. Line 119-131: it should be noted that plasma is not the best sample for detection of CDV genome. In blood the cell is mostly white blood cell-associated, other potential clinical specimens for diagnostic RT-PCR include swabs of nose (also mentioned in line 333!), throat or rectum or a urine sample.
8. Lines 218-220: the results section is not easy to read. I would propose to move paragraph 2.1 (which does not include materials or methods) to the start of the results.
9. Line 225/226: I do not agree with the statement: distemper in zoos have been reported earlier, and raccoons have been implemented ads potential source of the infection (ref 19).
10. Lines 226-235: this is discussion / speculation, not a result. This should be moved to the discussion. The authors should also consider the possibility of free-ranging small carnivores (eg mustelids) as potential source of the infection.
11. Line 237.238: the qRT-PCR was performed on plasma, not on whole blood.
12. Line 242: the legend to table 1 should at least include description that these are data obtained from plasma samples. It is not clear to me why Ct values below 11 are considered negative?
13. Lines 245-246: delete this sentence.
14. Line 268: figures 1 and 2 can be combined into one figure with two panels. Similarly, figures 3-6 can be combined into one figure with four panels.
15. Line 292: reference 5 is a study on adenovirus, not morbillivirus. CDV does not cause disease in domestic cats, and this sentence is thus incorrect.
16. Results: the qRT-PCR data reported in table 1 are the only evidence that the five surviving raccoons were also infected. The authors should have tested these animals for CDV-specific antibodies after convalescence. In addition, they should have attempted to obtain a (partial) sequence of the virus.
17. Line 308: included the disease (distemper) in this sentence.
18. Line 310: CDV is the abbreviation of the virus, not of the disease.
19. Line 399: why was IRB approval required for this study? It is primarily a diagnostic study, without an experimental component.
The manuscript should be edited for deleted spaces, e.g. line 44 (includes18), line 54 (Itaffects), lines 63 (recoveredanimals), etc.
Author Response
Dear Reviewers,
We would like to extend our gratitude for the careful reviews and for very constructive suggestions! Please find attached the Cover Letter / Response to reviewer containing all the points addressed one by one. We hope the paper now meets your expectations.
Kind regards,
Ioan Hutu

Reviewer 3 Report
Dear Authors, the article has the advantage of interesting contents, but the linguistic structure of the article makes the reader lose its scientific relevance. A major overhaul of English is of utmost importance and absolutely necessary.
Quality of English language must absolutely be improved.
Author Response

(The authors gave the same response as above.)

Reviewer 4 Report
1. Overview and general recommendation:
In this case report, Authors described the occurrence of a canine distemper virus infection in raccoons hosted in a zoo in Romania. As limited data on zoo and/or wild animals with CDV infection are currently available, these data deserve interest, but some descriptive limits were observed. I suggest an overall and wide review of this manuscript. I included below some suggestions and comments to support the Authors to improve the description throughout the manuscript.
Major comments:
- Title: Despite a case of CDV infection in raccoons was described, the title introduces to the description of a different and conflictual topic (“similarities” and “interspecies transmission”) and to consider these raccoons as wild animals: several similarities were observed between the lesions in these raccoons and those commonly described in domestic dogs, but the interspecies transmission was not demonstrated. This study, reported as a case report, is already a reasonable novelty, but without a clear connection with other species. Moreover, despite commonly considered as wild animals, this case report describes the CDV infection in zoo hosted animals. I suggest to carefully take in account these considerations throughout your revision.
- Overall, this study deserves a wide scientific review, an editing of the whole text, and a wide English language review. Data are interesting but they deserve to be better described.
- Canine distemper virus is not a disease but a viral agent cause of the infection commonly well notes as distemper. I suggest to review this detail in the manuscript. Moreover, I suggest to carefully consider only CDV in this study and not generically other members of Morbillivirus genus (i.e., feline morbillivirus and other ones).
- Lines 66-75: Direct transmission is more common that indirect transmission, also considering the lability of this virus. Moreover, the “easy” transmission between domestic and wild species was described in specific contexts. This part should be revised and synthesized.
- Lines 83-84: This sentence should be carefully revised and, if available, a reference should be included.
- Par.2.6: Why lung samples were collected (line 161) but these were not analysed?
- Results section: in this section, results were intermingled with comments (i.e., lines 223-224; 226-235; 245-246; 278-279; 299) that could be moved or considered in the following section.
- Par.3.3: pharyngeal ulcers are not easily referable to the CDV infection or a “septicemic evolution”, but these lesions were not documented or further analysed. Septicaemia is commonly referred to a serious bloodstream infection due to a bacterial infection, therefore this hypothesis should be better addressed.
- Lines 270-274: this data suddenly appears in this part of the section and should be included in the correct sections. According to gross lesions and sudden death, CPV-2 infection was the main pathogen to detect and should be included in the differential diagnosis.
- Discussion section: this is the paragraph that deserves more attention and needs a wide review. As above observed, two main conclusions (comparison and potential transmission between dogs and raccoons) should be more critically reconsidered. Moreover, are Authors sure that raccoons could represent a threat to dogs or vice versa?
Minor comments:
- Lines 15 and 25: “sometimes” should be revised.
- Lines 16 and 26: I suggest to include “, Romania” just after “Timisoara Zoo”; please check if “TimiÈ™oara Zoological Garden (TimiÈ™oara Zoo)” should be preferred.
- Line 17: “in raccoons” could be removed.
- Line 19: “all of which” is not correct if referred to all previous cited tests.
- Lines 21-23: this is a generic conclusion (please, see at first major comment) and is not a conclusion for this specific study. I suggest to revise it.
- Line 27: “was accomplished” should be revised.
- Line 28: “repeated” should be revised.
- Lines 30-32: this description should be revised for this section.
- Lines 36-39: this is a generic conclusion (please, see at first major comment) and is not a conclusion for this specific study. I suggest to revise it.
- Keywords: I suggest to include more keywords, to support the indexing of this manuscript.
- Lines 44-50: please, check if the term “raccoon” should be referred only to “Procyon lotor” and include some data on raccoons in Romania.
- Line 51: “The disease”: which one? Please, move “enveloped” before “single-stranded”; “from the family of” should be revised.
- Line 53: a threat to veterinarians and owners or to animals?
- Line 56: “family’s” should be revised.
- Lines 56-59: as in a major comment, I suggest to carefully review the host range for CDV, including clear specific references, and avoiding “etc”.
- Lines 61-62: “status” should be revised; all cited studies were limited to the last forty/fifty years, not to the last century.
- Lines 64-65: this sentence should be revised and, eventually, a reference included.
- Line 82: “are dependent on” should be revised and, if available, a reference should be included.
- Lines 95-96: please, revise this sentence.
- Line 101: “caused by the interspecific infection with CDV”?
- Lines 104-105: this part could be removed.
- Line 106: any approval code was provided?
- Lines 108-109: please, provide the total number of raccoons hosted in this zoo and when veterinarian first reported the clinical signs.
- Line 110: “(Procyon lotor)” should be moved at line 108 or removed.
- Line 111: “up to”?
- Lines 111-112: these are clinical signs common to CDV infection but also to other microbiological infections or to other causes.
- Line 114: “on ice”? it is haemolytic for EDTA blood samples. Perhaps, at +4°C?
- Lines 117-118: “and” could be removed; “2 and 3 days after the onset of clinical signs” could be preferred.
- Line 121: “Total” could be preferred to “Viral”.
- Line 124: “commercial” could replace “in vitro diagnostic”; add a hyphen in “One Step”.
- Line 125: I suggest to add “Real Time PCR” after “qualitative”
- Lines 127-128: please, specify that controls were provided in the kit by the manufacturer.
- Lines 142-144: To “both dead raccoons,” could be removed because redundant.
- Lines 146-147: A few additional details could be included.
- Line 148: “mycosis” should be revised.
- Line 149: “inoculations” is not a proper term.
- Line 151: “various media” is too generic; the following part of the sentence “,depending..” could be removed if more details are included.
- Lines 154-155: this part could be removed.
- Line 166: please, include details for the coniugate used.
- Lines 171-194: this part could be synthesized.
- Line 197: please, add “, Romania,” after “Bucharest”.
- Line 222: why this reference? Reference in the results section should be avoided unless necessary. Are Authors sure that “body growth status” is correct?
- Lines 224-226: these are information that should be included in the M/M section.
- Table 1: I suggest to include an additional column, to include clinical signs/outcome information.
- Lines 242-244: I suggest to remove this part or to include only the positive Ct range.
- Par.3.4: this paragraph includes a repetition of the M/M section and, therefore, should be rephrased.
- Par.3.5: Tissue collected for these examinations should be detailed in the M/M section and not suddenly included in this paragraph. Bacteriological examination was limited to E. coli detection?
- Figures 1 and 2: these images are not suggestive of the very limited description in the captions and in the related paragraph.
- Lines 277-278: this part should be included in the correct section.
- Lines 279-281: this part suddenly appears and is not clear.
- Lines 282-287: this part should be included in the M/M section.
- Lines 291-293: this part should be better addressed.
- Lines 313-323: this part is not related to the observed lesions/obtained results and, therefore, these conditions cannot be linked.
- Line 324: I suggest to replace “special” with “specific”;
- Line 325: I suggest to replace “symptoms” with “clinical signs”.
- Line 329: Are Authors sure?
- Lines 333-338: Authors should include more references and some data should be revised.
- Lines 342-345: this part conflicts with those reported at line 253.
- Lines 387-389: the “method to diagnose CDV” infection “involves performing PCR assay on...unvaccinated puppies”?
English language should be improved after a wide revision of the manuscript.
Author Response

(The authors gave the same response as above.)

Round 2
Reviewer 1 Report
In accordance with the reviewers' recommendations, the manuscript was improved. However, I also agree with reviewer 2 (i.e. non-free-ranging raccoons "wild raccoons" should be changed to captive raccoons).
Author Response
Dear Editors,
We would like to thank the reviewers for their pertinent suggestions, which were much appreciated. Please find attached detailed explanations for all the points raised. We hope we have rightly addressed all of them and that the paper now meets all expectations.
Kind regards,
Ioan Hutu

Reviewer 2 Report
The authors have adequately addressed most of the review comments. However, I still do not understand why the title refers to "wild raccoons", as they outbreak took place in a zoo I think this should be changed to "captive raccoons".
NA
Author Response

(The authors gave the same response as above.)

Reviewer 3 Report
Dear Authors, the draft paper can have a scientific interest as case report in zoo's raccoons. Currently, the paper needs major revision in order to improve the scientific quality and soundness, which I will list point by point:
Abstract
Line 28 - Please, use the word immunohistochemistry and histopathology instead of immunohistochemical analysis and histopathological analysis.
Line 29 - Please, rephrase the sentence - "which confirmed the infection" with "which confirmed CDV infection".
line 30 - Please, rephrase the sentence - "were observed clinically" with "were observed".
Lines 31-32 Please, rephrase the sentence "Immunohistochemical analysis allowed the fixation of the primary antibodies on the viral neurocapsid" with "Immunohistochemistry has immunolabeled CDV nucleocapsid antigen"
Line 32 - Please, use the word Histopathology instead Histopathological analysis;
Line 33 - Please, refine the sentence "revealed lymphocyte depletion". Indicate if the lymphocyte depletion occur in GALT or Mesenterial lymph-nodes.
Line 33 Please, refine the sentence "inclusion in the intestine". Indicate the type of cell with the included bodies. Could it be intestinal mucosal epithelial cells? Which part of the intestine is affected: small or large?
Lines 35 - 36 Please rephrase the sentence for improving the concept about of similarity of lesions, gross and histo, found in the dog and in the raccoon.
Line 39 - Please, delete the word "exam" after IHC.
Key words
Please, use the word immunohistochemistry and histopathology instead of immunohistochemical analysis and histopathological analysis.
no
Author Response

(The authors gave the same response as above.)

Reviewer 4 Report
I reviewed once again this manuscript but, except for including most of reviewers suggestions, the scientific soundness of this manuscript was not improved. I suggest to reconsider this manuscript, overcoming the main limits as widely already observed in the previous stage of the review. Such long description for the limited results and some not specific conclusions are preventing a positive evaluation of this manuscript for this Journal.
Best regards,
Author Response

(The authors gave the same response as above.)

Round 3
Reviewer 3 Report
Dear Authors,
in general, the text must be perfected by putting the spaces between the words which in many cases are missing. I must point out that the plural of zoo is zoo. If you want to use the plural to better highlight your concept, I suggest you use the form "zoological parks".
In detail:
The introduction, very detailed, is however too long. Parts of it can be taken out and used in the discussion.
The "passive vectors" can be identified with a specific word i "fomites" (objects or materials which are likely to carry infection, such as clothes, utensils, and furniture) (line 71). I advise you to specify which are the cells, of the respiratory and digestive systems, where the CDV virus replicates (epithelial cells)( line 83).
It is also useful to highlight lesions involving the tooth enamel (line 97).
Line 102 - Please, change "was conducted" with "was performed";
Line 103 - 104 Please, replace "For this purpose a series of " with "Ancillary";
The chapter of "2. Materials and methods" should be rewritten by highlighting that it is not a "case" but an outbreak of CDV.
Paragraph "2.2 Detection of canine ..."
Line 135 - Please, remove the word "particles" and replace it with "antigen";
Line 141 - Please, remove the word instruction and replace with "data sheet protocol".
Paragraph "2.4 Toxicological...."
Line 158 Plaese, rewrite intoxication as intoxication/poisoning;
Paragraph "2.5 Microbiological..."
Line 176 - Please, replace the word "fragments" with "segments";
Pargraph "2.6 - Histopathological and immunohistochemical examination" replace the title with "Histopathology and immunohistochemistry"
Line 192 - Please, replace "histopathological examination" with "histopathology";
Line 222. Please, specify which kind of image capture system has been used associated to CX41 Olympus microscope;
Paragraph "2.7 Electron microscopic examination" in the test put in evidence that is a "Transmission Electronic Microscope - TEM" investigation.
Please, specify the the technical details of TEM. Manufacturer and model.
Line 230-232 "The smaller ... resolution" I think that it is not neccessary to specify it. Please remove the sentence.
Paragraph "3.3 Necropsy exam"
Line 260-262 I advise you to rewrite the sentence as follows to give it greater communicative/explanatory effectiveness.
"Gross pathology was characterized by ..."
Line 262 - 264 "The severity...evolution" the sentence is not a results but it is a conclusion, so I advise you to remove from this paragraph. However, in the study no investigations on immune status were performed (e.g. flow cytometry), therefore your conclusion is not supported by scientific data. You can cite it in the discussion if supported by a reference.
Line 264 Please, rephrase the sentence in "In the lumen of small intestine adult nematodes of the genus Ascaris app. were recognized".
Paragraph "3.5 Microbiological exam"
From a purely microbiological point of view you have investigated only for bacteria and fungi, while you have ascertained the identification of the presence of viral particles with the TEM investigation. TEM is a morphological and not a microbiological investigation, therefore I advise you to remove the TEM microphotograph and comments from the microbiological results and include in paragraph on the TEM (Paragraph 3.7).
Line 277 - Please "morphopathological lesions" choice one of the because both words have the same meaning, therefore the double use is pleonastic.
Pargraph "3.6 Histopathological and immunohistochemical examination"
Please change the title in "Histopathology and immunohistochemistry"
Line 284 - Please rewrite the sentence as follow: intestine mucosa has shown necrosis (diffuse or focal? please specify as well if it is transmural of localized only in the mucosa.
Line 287 - Please, rephrase the sentence because inclusion body are located inside the cell (nucleus or cytoplasms), while indicate the organ (brain) is not correct.
Line 289 - Please, rephrase the sentence as follow "Immunohistochemistry of intestine (please indicate the tract of the intestine - small or large; better the intestine tract e.g duodenum, etc.) has shown immunopositivity ..."
Lines 291-294 The inclusion are not produced by morbilliviruses but are their aggregates visibile at optical microscope. I invite to rewrite the sentences.
Paragraph "4. Discussion"
First of all I invite the authors to use the impersonal verbal form.
The text setting of the paragraph "4. Discussion" is too broad and sometimes appears dispersive. Careful focusing is recommended.
Pargraph "5. Conclusion"
Line 422 - Please, change "in wild or domestic areas" with "in nature or urban areas".
Lines 430-433 Please, rewrite the sentence for improving the communicative effectiveness in consideration that it is the last sentence of the text that usually remains imprinted in the reader.
Dear Authors, a minor editing is required as indicate in "Comments and suggestion" box.
Author Response
Cover Letter
Thank you for the comments, we addressed all the issues and made the required changes, as follows:
In general, the text must be perfected by putting the spaces between the words which in many cases are missing. I must point out that the plural of zoo is zoo. If you want to use the plural to better highlight your concept, I suggest you use the form "zoological parks".
- The words do not appear connected in the version we edited, there must be some issue when we uploaded, or the reviewer downloaded the paper. We hope it is now correct.
- We replaced “zoos” with “zoological parks”.
The introduction, very detailed, is however too long. Parts of it can be taken out and used in the discussion.
- Parts of the introduction were taken out and used in the discussion.
The "passive vectors" can be identified with a specific word i "fomites" (objects or materials which are likely to carry infection, such as clothes, utensils, and furniture) (line 71). I advise you to specify which are the cells, of the respiratory and digestive systems, where the CDV virus replicates (epithelial cells)( line 83).
- We replaced "passive vectors" with the word "fomites".
- We specified which the cells are and we move the paragraph to discussions.
It is also useful to highlight lesions involving the tooth enamel (line 97).
- We highlighted lesions involving the tooth enamel and we move the paragraph to discussions.
Line 102 - Please, change "was conducted" with "was performed";
- We replaced the word “conducted” in 3 instances.
Line 103 - 104 Please, replace "For this purpose a series of " with "Ancillary";
- We made the change.
The chapter of "2. Materials and methods" should be rewritten by highlighting that it is not a "case" but an outbreak of CDV.
- We performed several modifications, we replaced “case study” with “outbreak of CDV”.
Paragraph "2.2 Detection of canine ..."
- We change the title in “Detection by RT-qPCR”
Line 135 - Please, remove the word "particles" and replace it with "antigen";
- We made the change.
Line 141 - Please, remove the word instruction and replace with "data sheet protocol".
- We replaced it.
Paragraph "2.4 Toxicological...."
- We replaced “Toxicological screening” with “Toxicology.”
Line 158 Please, rewrite intoxication as intoxication/poisoning;
- We added ‘’poisoning’’.
Paragraph "2.5 Microbiological..."
- We replaced “Microbiological screening” with “Microbiology”.
Line 176 - Please, replace the word "fragments" with "segments";
- We replaced it in all the instances.
Pargraph "2.6 - Histopathological and immunohistochemical examination" replace the title with "Histopathology and immunohistochemistry"
- We replaced both in all instances.
Line 192 - Please, replace "histopathological examination" with "histopathology";
- We replaced it.
Line 222. Please, specify which kind of image capture system has been used associated to CX41 Olympus microscope;
- The microscope is with 3MP CMOS Digital Color Camera.
Paragraph "2.7 Electron microscopic examination" in the test put in evidence that is a "Transmission Electronic Microscope - TEM" investigation.
- We added “TEM investigation”, to be more specific.
Please, specify the technical details of TEM. Manufacturer and model.
- We specify the type of the Hitachi TEM.
Line 230-232 "The smaller ... resolution" I think that it is not neccessary to specify it. Please remove the sentence.
- We removed this sentence: “The smaller the electron-dense substance used, the deeper it penetrates the surface structures of the virus, rendering higher resolution.”
Paragraph "3.3 Necropsy exam"
- We changed it to “Necropsy.”
Line 260-262 I advise you to rewrite the sentence as follows to give it greater communicative/explanatory effectiveness.
"Gross pathology was characterized by ..."
- We rephrased it.
Line 262 - 264 "The severity...evolution" the sentence is not a results but it is a conclusion, so I advise you to remove from this paragraph. However, in the study no investigations on immune status were performed (e.g. flow cytometry), therefore your conclusion is not supported by scientific data. You can cite it in the discussion if supported by a reference.
The paragraph was changed with: The severity of detected anatomo-pathology and clinical signs in the hosts was associated with the presence of secondary infections with E. coli
Line 264 Please, rephrase the sentence in "In the lumen of small intestine adult nematodes of the genus Ascaris app. were recognized".
- We rephrased it.
Paragraph "3.5 Microbiological exam"
- We replaced it with “Microbiology.”
From a purely microbiological point of view you have investigated only for bacteria and fungi, while you have ascertained the identification of the presence of viral particles with the TEM investigation. TEM is a morphological and not a microbiological investigation, therefore I advise you to remove the TEM microphotograph and comments from the microbiological results and include in paragraph on the TEM (Paragraph 3.7).
We changed it
Line 277 - Please "morphopathological lesions" choice one of the because both words have the same meaning, therefore the double use is pleonastic.
- We replaced pleonastic phrases in all instances.
Pargraph "3.6 Histopathological and immunohistochemical examination"
- We replaced them in all instances.
Please change the title in "Histopathology and immunohistochemistry"
- We changed it.
Line 284 - Please rewrite the sentence as follow: intestine mucosa has shown necrosis (diffuse or focal? please specify as well if it is transmural of localized only in the mucosa.
- We changed the sentence “The histopathology of intestine samples evidenced necrosis.” => “Intestine mucosa has shown diffuse necrosis localized in the mucosa”
Line 287 - Please, rephrase the sentence because inclusion body are located inside the cell (nucleus or cytoplasms), while indicate the organ (brain) is not correct.
- We replaced “Brain inclusions” with “Inclusions at the level of encephalic cells”
Line 289 - Please, rephrase the sentence as follow "Immunohistochemistry of intestine (please indicate the tract of the intestine - small or large; better the intestine tract e.g duodenum, etc.) has shown immunopositivity ..."
- We made the modification, as follows: “Immunohistochemistry of small intestine has shown immunopositivity, …”
. Lines 291-294 The inclusion are not produced by morbilliviruses but are their aggregates visibile at optical microscope. I invite to rewrite the sentences
- We corrected this erroneous phrasing.
Paragraph "4. Discussion"
First of all I invite the authors to use the impersonal verbal form.
- We made several modifications to address the formality aspect, e.g.: “we utilized electron microscopy” was modified: “electron microscopy was utilized”, etc.
The text setting of the paragraph "4. Discussion" is too broad and sometimes appears dispersive. Careful focusing is recommended.
We made it less dispersive by focusing more on the main issues under scrutiny…..
Pargraph "5. Conclusion"
Line 422 - Please, change "in wild or domestic areas" with "in nature or urban areas".
- We had already removed the phrase in a previous version.
Lines 430-433 Please, rewrite the sentence for improving the communicative effectiveness in consideration that it is the last sentence of the text that usually remains imprinted in the reader.
- We modified the last phrase:
“In conclusion, zoological parks may help reduce CVD outbreaks by: minimizing contact with infected animals, interdicting visitors’ pets, controlling wild host animals, and vaccinating captive animals which may be carriers of CDV.”

Reviewer 4 Report
I thank Authors for their replies. At this stage of the review, I have none further comment than previous ones
Quality of English language and spelling could be improved
Author Response
In our view, the aim of the paper was essentially to present a case occurring in a specific area for isolated/captive animals rather than to present new epidemiological aspects or biomolecular analysis. We therefore maintain that our aims were largely met, as initially projected. As suggested (and modified) in the title, it remains mainly a case study on Unusual canine distemper virus infection in captive raccoons.
